# Survival Outcomes in the Canadian Merkel Cell Carcinoma Population Between 2000 and 2018 and Descriptive Comparison with the American Joint Committee on Cancer 8th Edition Staging System—A Study from the Pan-Canadian Merkel Cell Collaborative

**DOI:** 10.3390/cancers17193238

**Published:** 2025-10-06

**Authors:** Brittany Dingley, Megan Delisle, Anne Light, Sameer Apte, Ranjeeta Mallick, Trevor Hamilton, Heather Stuart, Martha Talbot, Gregory McKinnon, Evan Jost, Eva Thiboutot, Valerie Francescutti, Salsabila Samman, Alexandra M. Easson, Angela Schellenberg, Shaila Merchant, Julie La, Kaitlin Vanderbeck, Frances C. Wright, David Berger-Richardson, Pamela Hebbard, Olivia Hershorn, Rami Younan, Erica Patocskai, Samuel Rodriguez-Qizilbash, Ari Meguerditchian, Vanina Tchuente, Suzanne Kazandjian, Alex Mathieson, Farisa Hossain, Jessika Hetu, Stephanie Johnson-Obaseki, Carolyn Nessim

**Affiliations:** 1Division of General Surgery, The Ottawa Hospital, Ottawa, ON K1H 8L6, Canadamegan.delisle@umanitoba.ca (M.D.); sapte@toh.ca (S.A.); 2Division of General Surgery, Faculty of Medicine, University of Ottawa, Ottawa, ON K1N 6N5, Canada; stjohnson@toh.ca; 3Ottawa Hospital Research Institute, Ottawa, ON K1Y 4E9, Canada; 4Division of General Surgery, BC Cancer Agency, Vancouver, BC V5Z 4E6, Canada; trevor.hamilton@vch.ca (T.H.); heather.stuart@vch.ca (H.S.);; 5Division of General Surgery, University of Calgary, Calgary, AB T2N 1N4, Canada; greg.mckinnon@albertahealthservices.ca (G.M.); eajost@ucalgary.ca (E.J.);; 6Division of General Surgery, Hamilton Health Sciences, McMaster University, Hamilton, ON L8N 3Z5, Canada; francesv@hhsc.ca (V.F.); salsabila.samman@medportal.ca (S.S.); 7Division of General Surgery, Mount Sinai Hospital, Princess Margaret Cancer Center, University Health Network, Toronto, ON M5G 1X5, Canada; alexandra.easson@sinaihealth.ca (A.M.E.);; 8Division of General Surgery, Queen’s University, Kingston, ON K7L 3N6, Canada; shaila.merchant@kingstonhsc.ca (S.M.);; 9Department of Pathology and Molecular Medicine, Queen’s University, Kingston, ON K7L 3N6, Canada; 10Division of General Surgery, Sunnybrook Health Sciences Center, Toronto, ON M4N 3M5, Canada; 11Department of Surgical Oncology, CancerCare Manitoba, Winnipeg, MB R3E 0V9, Canada; phebbard@cancercare.mb.ca; 12Division of General Surgery, Manitoba Health Services, Winnipeg, MB R3B 3M9, Canada; 13Division of General Surgery, Centre Hospitalier de L’Université de Montréal, Montreal, QC H2X 0C1, Canada; 14Division of General Surgery, Unversity of Sherbrooke, Sherbrooke, QC J1H 5N4, Canada; jessika.hetu@usherbrooke.ca; 15Division of General Surgery, McGill University Health Network, Montreal, QC H4A 3J1, Canada; 16Division of General Surgery, Memorial University, St-John’s, NL A1C 5S7, Canada; 17Department of ENT, The Ottawa Hospital, Ottawa, ON K1H 8L6, Canada

**Keywords:** Merkel cell carcinoma, Canada, cohort study, epidemiology, treatment

## Abstract

**Simple Summary:**

Merkel cell carcinoma is an aggressive skin malignancy. Due to its rarity, there is limited data on survival outcomes, staging, and treatment options. This study aims to characterize the Merkel cell population in Canada and describe its survival outcomes, including overall survival, disease-free survival, and cancer-specific survival. A comparison is made between the survival outcomes in a Canadian population and those of a U.S. cohort, and the implications of differences between these groups and the relevance and application of a staging system are explored.

**Abstract:**

**Background/Objectives**: Merkel cell carcinoma (MCC) is an uncommon but aggressive skin malignancy with a rising incidence. Limited data exist on the survival of MCC patients in Canada. This study analyzes the survival of patients diagnosed with MCC in Canada between 2000 and 2018 compared to those reported by the American Joint Committee on Cancer (AJCC) 8th edition. Risk factors included in the database were sex, age, and immunosuppression. **Methods**: We conducted a multicenter retrospective cohort study including patients diagnosed with stage I–IV MCC aged ≥18 from 10 Canadian university centers and three provinces. We evaluated differences in survival compared to the cohort included in the AJCC 8th edition. **Results**: Among 899 patients diagnosed with MCC in Canada, 327 (36.4%) had stage I, 195 (21.7%) had stage II, 305 (33.9%) had stage III, and 72 (8.0%) had stage IV at presentation. When examining risk factors, 61.1% (549) were male, 10.2% (92) were immunosuppressed, and age at diagnosis was 75 years (±11). The five-year overall survival for patients diagnosed in Canada at stage I was 56.8%, stage IIA 54.0%, stage IIB 28.0%, stage IIIA 52.7%, stage IIIB 40.2%, and stage IV 13.9%. **Conclusions**: Survival from MCC is low in Canada across all stages. Compared to the AJCC 8th edition, patients diagnosed with MCC in Canada have similar survival rates, except for patients diagnosed with stage IIIB, who have lower survival rates in the AJCC 8th edition. Further research is needed to improve the survival of this rare malignancy.

## 1. Introduction

Merkel cell carcinoma (MCC) is a rare and aggressive cutaneous malignancy characterized by a poor prognosis [1,2]. Risk factors for MCC include immunosuppression [3], fair skin, Merkel cell polyomavirus (MCPyV) infection [4,5], UV-B light exposure, advancing age, and male sex [6]. The American Joint Committee on Cancer (AJCC) 8th edition staging can inform prognosis and guide treatment decisions in MCC. The AJCC 8th edition staging for MCC was derived using a cohort of patients from the National Cancer Data Base (NCDB), which contains information on more than 70% of newly diagnosed cancer cases in the United States [7]. There are differences in cancer management between the United States and Canada, which may be attributed to the health care system, cancer screening guidelines, availability and coverage of treatment options, clinical practice guidelines, access to care, and funding for cancer research [8,9]. Differences in geographic location may be relevant, evidenced by a variation in disease biology when comparing Canadian MCC vs. Australian MCC [10]. These differences can potentially affect patient outcomes, including survival and recurrence rates, due to variations in the quality and timeliness of care, as well as other factors such as patient characteristics and the stage at which cancer is diagnosed [9]. Given the differences between the United States and Canada, there is a need to understand if the current AJCC 8th edition staging system provides relevant prognostic information for patients diagnosed with MCC in Canada. To address this gap in knowledge, we conducted a retrospective cohort study of 899 patients diagnosed with MCC in Canada between 2000 and 2018 who were treated at 10 university centers and across three provinces. This study analyzes the survival of patients diagnosed with MCC in Canada and compares it to that reported by the AJCC 8th edition.

## 2. Materials and Methods

Ethical approval was obtained from the Research Ethics Board (REB) across all participating centers, and the study adhered to the STrengthening the Reporting of OBservational studies in Epidemiology (STROBE) guidelines [11]. Approval was obtained from the IRB of the Ottawa Hospital—Ottawa Hospital Research Institute. Approval code is 20160479-01H on 23 June 2016.

### 2.1. Study Design

This study is an observational, non-interventional, retrospective study.

### 2.2. Setting and Study Population

A multicenter database of MCC cases was created through collaboration with 13 Canadian university centers. Data were collected from 10 university centers and three provincial cancer registries.

### 2.3. Inclusion and Exclusion Criteria

The inclusion criteria for this study were patients ≥18 years old with MCC confirmed histologically on biopsy or surgical pathology between 2000 and 2018. Patients were excluded if they had an equivocal diagnosis on pathology, incomplete data for analysis, or were diagnosed with another malignancy within five years of diagnosis (excluding low-risk squamous cell and basal cell carcinoma).

### 2.4. Data Collection

Data were collected from multidisciplinary meeting notes, pathology databases, medical records (paper or electronic), vital statistics records, and pre-existing cancer databases (if available). Data were collected by experienced clinical research assistants and stored in a password-protected Excel database. Data transfer agreements were held between all participating centers and the Ottawa Hospital, where all analysis was completed. Data were de-identified and combined into a single electronic spreadsheet for data analysis at the coordinating center.

### 2.5. Data Variables

Data collected for the study included patient characteristics (date of birth, sex, treating center, Charlson comorbidity, immunosuppressive status), tumor characteristics (location, size, depth of invasion, nodal status, stage at diagnosis), management (wide local excision, sentinel lymph node biopsy, lymph node dissections, neoadjuvant and adjuvant treatments including radiation), and patient outcomes (recurrence, date of death, cause of death, date of last follow-up, vital status).

The primary outcome was five-year overall survival. Secondary outcomes included five-year disease-free survival (DFS) and cancer-specific survival (CSS). Survival outcomes were reported by disease extent using prognostic categories reported by the AJCC 8th edition to allow for comparison across cohorts. Survival outcomes for the cohort, including the AJCC 8th edition, were obtained from those previously published.

### 2.6. Statistical Analyses

Descriptive statistics were used to summarize the baseline characteristics of the study population. Continuous variables are reported as mean and standard deviation or median and interquartile range, while categorical variables are reported as counts and percentages. We used Kaplan–Meier curves to estimate the survival probability over time. The differences between the survival curves were tested using the log-rank test. To identify the independent predictors of the outcome, we performed a Cox proportional hazards regression analysis. The results of the Cox regression are presented as hazard ratios (HR) with 95% confidence intervals (CI).

## 3. Results

### 3.1. Canadian Patient Characteristics

A total of 899 patients were included in our study population, 512 (57.0%) from 10 Canadian University centers and 387 (43%) from 3 Canadian provincial registries between 2000 and 2018 (Table 1). Among the total study population, 61.1% (549) were male, 10.2% (92) were immunosuppressed, the age at diagnosis was 75 years (±11), 68.7% (618) had a Charlson Comorbidity Index ≥ 4, and 44.6% (401) were diagnosed with MCC on their head or neck. A total of 36.4% (327) had stage I, 21.7% (195) had stage II, 33.9% (305) had stage III, and 8.0% (72) had stage IV at presentation. Patients were treated with surgery plus radiation (353, 39.3%), surgery alone (282, 31.4%), radiation alone (171, 19.0%), or no treatment (71, 7.9%) (Table 1). No patients received immunotherapy, as this was not an option during this time period.

### 3.2. Overall Survival Among Canadian Patients

The median follow-up for overall survival was 1.9 (interquartile range 0.8–4.3) years. Among lymph node negative patients (N0), patients with lower T stage had a significantly improved overall survival (*p* < 0.01, Figure 1). Among patients with regional lymph node metastases, patients with a larger burden of lymph node metastases had a significantly worse overall survival (*p* < 0.01, Figure 2). However, patients with regional lymph node macrometastases and an unknown primary (N1b unknown primary) had similar survival to patients diagnosed with microscopic regional lymph node metastases (N1a). When categorized as local, nodal or metastatic disease, a larger proportion of patients with local disease were alive at five years (five-year OS 54.3%, 49.4–59.0%) compared to those with nodal (five-year OS 46.2%, 40.0–52.2%) and distant metastatic disease (five-year OS 13.9% 6.8–23.3, *p* < 0.01) (Table 2).

When compared by AJCC 8th edition substage (Figure 3), patients with stage IIB had a shorter survival (five-year OS 28%, 95% CI 7.7–53.1) compared to all other stages except for stage IV (five-year OS 13.9%, 95% CI 6.8–23.3), who had the lowest survival. Stage I and IIA had similar five-year OS at 56.8% (50.5–62.5) and 54% (45.1–62.0), respectively.

### 3.3. DFS and CSS Among Canadian Patients

Patients with stage IIB disease in the Canadian cohort experienced the shortest DFS compared to all other substages (Table 3). Stage I, IIA and IIIA had similar five-year DFS at 66.6% (60.2–72.2%), 58.7% (49.6–66.7%), and 63.9% (54.3–72%), respectively. For Stage IIIB, the five-year DFS is 40.0% (31.2–48.6%), which is similar to the five-year DFS in Stage IIB patients (five-year DFS 36.4%, 95% CI 11.2–62.9%). Five-year CSS showed similar trends to five-year DFS, with CSS being higher than DFS across all stages except for patients with stage IV disease, where five-year CSS and DFS were similar (Table 3).

### 3.4. Overall Survival Among Canadian Versus AJCC 8th Edition Patients

Among patients with no nodal disease, Canadian patients with T2/3 disease had a longer survival compared to the prognosis reported by the AJCC 8th edition, whereas survival was similar between cohorts for patients with T1 and T4 disease. In patients with regional nodal disease, the median five-year OS was longer for Canadian patients compared to the prognosis reported by the AJCC 8th edition, except for the N2/N3 group, which was similar between cohorts. Canadian patients with regional nodal metastatic disease had a longer survival compared to the AJCC 8th edition cohort. There was no clinically meaningful difference in overall survival between Canadian patients and the AJCC 8th edition among those with regional and distant metastatic disease. Compared to the AJCC 8th edition, Canadian patients with Stage IIIA and IIIB disease had longer survival, whereas survival was more similar for all other substages. For stage IIIA, the AJCC 8th edition five-year OS was 40.3 (37.5–43) compared to 52.7% (43.5–61.0) in the Canadian patients. For stage IIIB, the AJCC 8th edition five-year OS was 26.8% (23.4–30.4) compared to 40.2% (31.7–48.7) in the Canadian patients. For Stage I patients, there is a trend for slightly higher survival in the AJCC 8th edition (62.8%, 59.5–65.8) compared to Canadian patients (56.8%, 50.5–62.5%) (Table 2).

## 4. Discussion

This retrospective cohort study on MCC in the Canadian population, comparing outcomes to the AJCC 8th edition, underscores MCC’s aggressive nature and poor prognosis [1]. While the predicted five-year OS for all cancers in Canada is 64%, patients with MCC in our study faced a bleaker prognosis than those only having local disease, showing a five-year OS of 54.3% (49.4–59.0%). The study validates the relevance of the AJCC 8th edition in the Canadian context, highlighting similar survival trends.

When comparing the OS of our cohort to those of the AJCC 8th edition cohort, outcomes appear to be similar. Notable differences include a lower five-year OS in stage I patients in the Canadian cohort and a higher five-year OS for stage III patients in the Canadian cohort. Although the reason for this cannot be determined due to the nature of our study, factors like access to care, treatment modalities, patient characteristics, or geographical location may contribute to the divergence. It is also important to note that the AJCC 8th edition population had a larger sample size with 1536 stage IIIA patients and 929 stage IIIB patients, compared with 152 and 150, respectively, in the Canadian population, which makes comparison between the two groups challenging. Despite these differences, the overall similarity in survival between the cohorts shows that the AJCC staging system is a useful and accurate prognostic tool for patients with MCC in a Canadian context.

The inclusion of CSS alongside OS provides important insights into MCC prognosis. While the prognosis of MCC has always been reported to be low, it is also well recognized that older and more comorbid patients have a higher incidence of MCC. In our study, the average age at MCC diagnosis in Canada was 75, and 68.7% of patients had a Charlson Comorbidity Index of ≥4, suggesting that age and comorbidities contribute to the poor prognosis. OS has been frequently used as a primary clinical endpoint because it is objective, definite, easy to measure, and unlikely to have researcher bias [12]. In a setting where most patient deaths are disease-specific, OS may provide a clear picture of the disease trajectory [13]. When other-cause death is more common, the use of multiple end points allows us to more accurately assess the effect of disease on survival, as may be the case in MCC. In support of this, we observed that OS was lower than that of CSS.

Similar to the AJCC 8th edition, we observed higher survival in all Stage III patients compared to Stage IIB. Additional research into the reasons is required. Stage IIB patients may have had a lack of nodal evaluation and, therefore, been inappropriately assigned to Stage II, despite an inherently worse prognosis. A review of our data shows that Stage IIB has the highest proportion of unknown pathologic nodal status, supporting that lack of pathologic nodal evaluation and stage migration may be playing a role in the poor prognosis of this substage. Stage migration has been demonstrated in the use of sentinel lymph node biopsy in melanoma and breast cancer, where patients with lower-burden nodal disease are recategorized from node-negative to node-positive, thereby improving the survival of this group overall [14,15].

The presence of patients with an unknown primary in the stage IIIA prognostic group may be another explanation for improved survival compared to the IIB population. Our results demonstrate that patients with an unknown primary have better survival than those with a known primary and nodal disease. This was consistent among Canadian patients and those included in the AJCC 8th edition [7]. It is hypothesized that the improved survival among patients with unknown primary may be due to a stronger underlying immune response contributing to the regression of the primary lesion [16]. Further, being immunocompromised is a known risk factor for MCC, and immunosuppression results in an earlier age at onset, a more aggressive course, and a worse prognosis [17].

The finding that Stage IIB patients have significantly worse outcomes than Stage III patients does highlight one weakness of prognostic groups, where increasing stage does not necessarily correlate with worse outcomes. This is a finding that is also seen in other disease sites, such as melanoma and certain high-risk stage II colon cancers [18,19], and is relevant because prognostic groups typically dictate access to treatment. Earlier-stage cancers with worse prognoses may have delayed introduction to new therapies. This delay was seen in melanoma, where adjuvant immunotherapy has been the standard of care for resected stage III and IV patients for many years, whereas adjuvant therapy in high-risk stage II patients has been available only relatively recently [18,20,21]. Prognostic groups are also often used as a tool for healthcare providers to communicate a patient’s disease extent and expected trajectory. We still acknowledge the usefulness of the AJCC 8th edition staging system in patient care, but it is relevant to know its limitations and nuances, especially in patient discussions and treatment decisions.

### Limitations

We did not have access to all prognostic factors such as Merkel cell polyomavirus or tumor mutational burden, which would provide greater insight into our cohort and might help to explain any differences identified. There is an absence of data on other risk factors for MCC such as UV exposure, Fitzpatrick skin type, or latitudinal location, which have been shown to play a role [22,23]. We also rely on the accuracy of staging information to ensure patients correspond to their assigned prognostic group. Specifically, nodal staging requires either knowledge of clinically involved nodes or a sentinel node biopsy for pathologic confirmation. Sentinel lymph node biopsy was utilized to a greater extent in the later years of this study. As such, patients may have been inaccurately staged. While this may have an impact, it is reassuring that 80% of patients underwent staging imaging, limiting the degree of radiographically positive nodal disease that otherwise may have been missed. While most substages were equally represented in the cohort, there were significantly fewer Stage IIB patients. As such, any outcome measures may be more subject to chance than in the larger groups. Despite this, we see the same trends observed in the NCDB database, which suggests a reliable finding. Further, this is one of the largest series of MCC patients, which reflects the rarity of Stage IIB patients and the need for additional efforts to understand this subset of patients and their outcomes.

This study is limited by sample size, given the rarity of MCC. The Canadian cohort includes 899 patients compared with 9387 in the AJCC 8th edition cohort. Although it is worthwhile noting that this is the largest Canadian MCC cohort that has been published, there are limitations with a smaller sample size, including the risk of random variability and less generalizability.

The current mainstay of treatment in Canada includes surgery and radiation based on current guidelines [24,25,26,27]. Several recent studies have shown improved outcomes in patients with recurrent and metastatic MCC treated with immunotherapy [16,28,29,30,31]. These clinical trials demonstrating favorable results with immune checkpoint inhibitors, such as Pembrolizumab and Nivolumab, led to their inclusion in the 2018 National Comprehensive Cancer Network (NCCN) guidelines for the treatment of patients with unresectable or metastatic MCC [26]. There is also recent evidence for the use of neoadjuvant immunotherapy in patients with resectable MCC [32]. We acknowledge that this is an older cohort before the era of immunotherapy, as was the AJCC 8th edition cohort, which was published in 2016, and Avelumab was only approved in 2017. No patients in the Canadian MCC population received immunotherapy, as this was not available at the time. As the uptake and implementation of immunotherapy in Canada increase, it is possible that these trends in survival may change. This limitation may make our study less applicable due to the emergence of new therapies, but it will serve as a baseline to compare outcomes after integrating these therapies into practice.

## 5. Conclusions

We observed similar patterns in prognosis between Canadian patients diagnosed with MCC compared to those reported by the AJCC 8th edition for OS. The largest difference was found in those with Stage III disease, where the Canadian cohort had higher survival than the AJCC 8th edition cohort. Despite these differences, the overall similarity in survival outcomes shows that the AJCC staging system is a useful prognostic tool in a Canadian context. The addition of secondary survival outcomes showed that, despite the poor prognosis associated with MCC, CSS was higher than OS, suggesting that patients diagnosed with MCC are dying from other causes. There is an urgent need for ongoing research into new therapies and improved access to emerging therapies to continue improving survival in this rare malignancy.

## Figures and Tables

**Figure 1 cancers-17-03238-f001:**
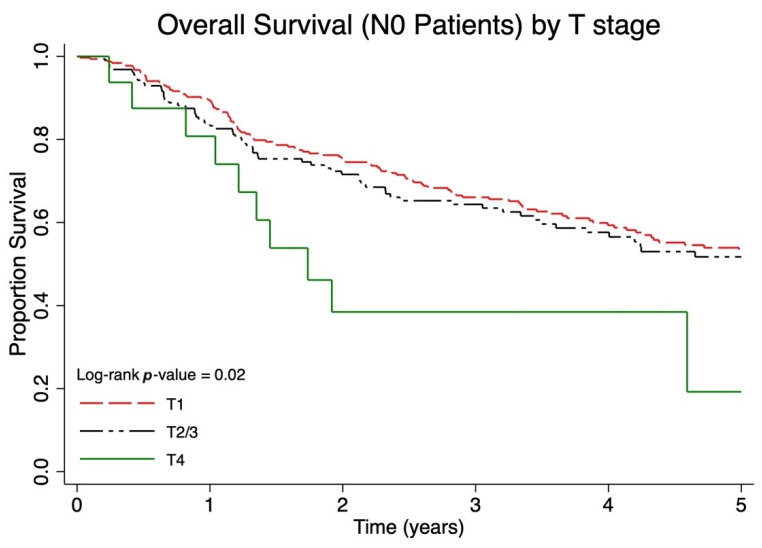
Five-year overall survival (OS) of Canadian Merkel cell carcinoma (MCC) patients presenting with local disease only stratified by primary tumor size using T categories (T1: primary tumor ≤ 2 cm, T2/3: primary tumor > 2 cm, T4: primary tumor invades fascia, muscle, cartilage, or bone).

**Figure 2 cancers-17-03238-f002:**
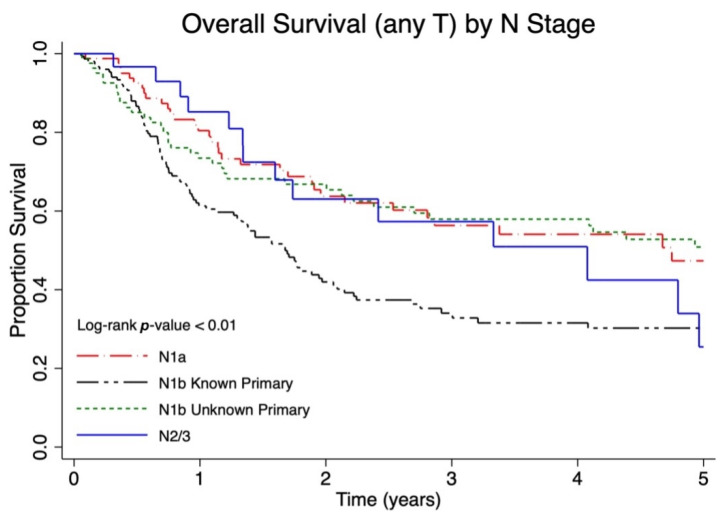
Five-year overall survival (OS) of Canadian Merkel cell carcinoma (MCC) patients presenting with nodal metastases stratified by occult nodal disease (N1a), clinically detected nodal disease with known primary tumor (N1b with known primary), clinically detected nodal disease with unknown primary tumor (N1b with unknown primary), and in-transit metastasis (N2/3).

**Figure 3 cancers-17-03238-f003:**
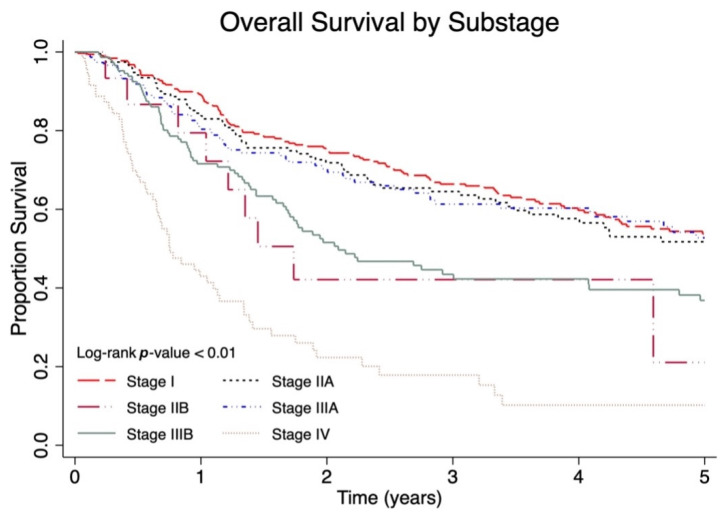
Five-year overall survival (OS) of Canadian Merkel cell carcinoma (MCC) patients stratified by American Joint Committee on Cancer (AJCC) 8th edition pathological substage including stage I (T1 N0 M0), stage IIA (T2/3 N0 M0), stage IIB (T4 N0 M0), stage IIIA (T1–4 N1a M0 and T0 N1b M0), stage IIIB (T1–4 N1b M0 and T0–4 N2 M0), and stage IV (T any Nany M1).

**Table 1 cancers-17-03238-t001:** Patient, Cancer and Treatment Characteristics of the Canadian Cohort.

	**Total**	
	N = 899	
Sex		
Male	61.1% (549)	
Female	38.8% (349)	
Missing	0.1% (1)	
Age at diagnosis (mean ± sd)		
	75 (11)	
Charlson Comorbidity Index		
0–1	6.6% (59)	
2–3	23.4% (210)	
≥4	68.7% (618)	
Missing	1.3% (12)	
Immunosuppressed		
No	87.1% (783)	
Yes	10.2% (92)	
Missing	2.7% (24)	
Geography		*p* value
East	57.0% (512)	<0.001
Central	20.4% (183)	<0.001
West	22.7% (204)	<0.001
Hospital		*p* value
TOH	9.2% (83)	<0.001
CHUM	4.3% (39)	<0.001
Sunnybrook	5.8% (52)	<0.001
PMH/UHN/Mount Sinai	11.6% (104)	<0.001
Memorial	2.3% (21)	<0.001
Manitoba	5.6% (50)	<0.001
Alberta	14.8% (133)	<0.001
Sherbrooke	1.7% (15)	<0.001
BC	22.7% (204)	<0.001
Queen*’*s	6.7% (60)	<0.001
McMaster	11.7% (105)	<0.001
McGill	3.7% (33)	<0.001
Site of Primary Tumour		*p* value
Head & neck	44.6% (401)	<0.001
Upper limbs/shoulder	21.2% (191)	<0.001
Lower limbs/hips	16.7% (150)	<0.001
Trunk/back	7.3% (66)	<0.001
Unknown primary	9.8% (88)	<0.001
Missing	0.3% (3)	<0.001
Stage at Diagnosis		*p* value
Stage 1	36.4% (327)	<0.001
Stage 2	21.7% (195)	<0.001
Stage 3	33.9% (305)	<0.001
Stage 4	8.0% (72)	<0.001
Substage at Diagnosis		*p* value
Stage 1	36.4% (327)	<0.001
Stage 2A	19.0% (171)	<0.001
Stage 2B	1.7% (15)	<0.001
Stage 3A	16.9% (152)	<0.001
Stage 3B	16.7% (150)	<0.001
Stage 4	8.0% (72)	<0.001
Missing	1.3% (12)	<0.001
T Stage at Diagnosis		*p* value
T0	9.8% (88)	<0.001
T1	45.3% (407)	<0.001
T2	27.0% (243)	<0.001
T3	7.9% (71)	<0.001
T4	4.9% (44)	<0.001
Missing	5.1% (46)	<0.001
N Stage at Diagnosis		*p* value
N0 (clinical or pathological)	59.2% (532)	<0.001
N1a	9.2% (83)	<0.001
N1b	26.1% (235)	<0.001
N2/3	3.4% (31)	<0.001
Missing	2.0% (18)	<0.001
Treatment Received		
None	7.9% (71)	
Surgery alone	31.4% (282)	
Surgery + radiation	39.3% (353)	
Radiation alone	19.0% (171)	
Other 2.4%	−22	
Median Follow-Up Period (years) (IQR)		
	1.9 (0.8–4.3)	

Abbreviations: sd, standard deviation; CHUM, Centre hospitalier de l’Université de Montréal; PMH, Princess Margaret Hospital; UHN, University Health Network; IQR, Interquartile range.

**Table 2 cancers-17-03238-t002:** Five-Year Overall Survival Canadian Population versus AJCC 8th Edition Population.

	Canadian Population			AJCC 8th Edition Population	
	Deaths/Total Population	5-Year OS	95% CI	5-Year OS	95% CI
Overall survival (N0 Population) by T stage					
T1N0	112/324	56.4%	50.1–62.2%	55.6%	54.1–57.5%
T2/3N0	62/171	53.9%	45.1–62.0%	41.1%	38.8–43.7%
T4N0	10/16	25.9%	7.2–50.0%	31.8%	24.8–38.9%
Overall survival (any T) by N stage					
T anyN1a	33/82	48.4%	35.8–59%	39.7%	36.7–42.7%
T anyN1b known primary	89/153	32.6%	24.8–40.6%	26.8%	23.3–30.4%
T0N1b unknown primary	36/82	49.7%	37.6–60.6%	42.2%	35.7–48.5%
N2/3	14/30	40.4%	21.1–59.1%	41.4%	25.0–57.0%
Overall survival by disease extent					
Local disease	59/188	54.3%	49.4–59.0%	50.6%	49.2–52.0%
Nodal disease	135/304	46.2%	40.0–52.2%	35.4%	33.0–37.6%
Metastatic disease	56/71	13.9%	6.8–23.3%	13.5%	11.0–16.3%
Overall survival by substage					
Stage I	112/326	56.8%	50.5–62.5%	62.8%	59.6–65.8%
Stage IIA	61/169	54.0%	45.1–62.0%	54.6%	49.3–59.7%
Stage IIB	9/15	28.0%	7.7–53.1%	34.8%	25.6–44.1%
Stage IIIA	58/151	52.7%	43.5–61.0%	40.3%	37.5–43%
Stage IIIB	75/150	40.2%	31.7–48.7%	26.8%	23.4–30.4%
Stage IV	56/71	13.9%	6.8–23.3%	13.5%	11.0–16.3%

Abbreviations: AJCC, American Joint Committee on Cancer; OS, overall survival; CI, confidence interval.

**Table 3 cancers-17-03238-t003:** Five-Year Overall Survival, Disease-Free Survival, and Cancer-Specific Survival in the Canadian Population.

	OS			DFS			CSS		
	Deaths/ Total Population	5-Year OS	95% CI	Deaths/ Total Population	5-Year DFS	95% CI	Deaths/ Total Population	5-Year CSS	95% CI
Stage I	112/326	56.8%	50.5–62.5%	79/326	66.6%	60.2–72.2%	18/326	91.5%	86.9–94.6%
Stage IIA	61/169	54.0%	45.1–62.0%	52/169	58.7%	49.6–66.7%	15/169	86.3%	78.0–91.5%
Stage IIB	9/15	28.0%	7.7–53.1%	7/15	36.4%	11.2–62.9%	2/15	77.8%	36.5–93.9%
Stage IIIA	58/151	52.7%	43.5–61.0%	41/151	63.9%	54.3–72.0%	21/151	79.8%	70.7–86.3%
Stage IIIB	75/150	40.2%	31.7–48.7%	72/150	40.0%	31.2–48.6%	34/150	65.3%	55.4–73.5%
Stage IV	56/71	13.9%	6.8–23.3%	31/71	41.0%	27.9–53.8%	27/71	46.5%	32.4–59.5%

Abbreviations: OS, overall survival; DFS, disease-free survival; CSS, cancer-specific survival; CI, confidence interval.

## Data Availability

The authors elect not to share data.

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
