# Peer review of "Survival Outcomes in the Canadian Merkel Cell Carcinoma Population Between 2000 and 2018 and Descriptive Comparison with the American Joint Committee on Cancer 8th Edition Staging System—A Study from the Pan-Canadian Merkel Cell Collaborative"

_cancers, 2025, doi:10.3390/cancers17193238_

Round 1

Reviewer 1 Report

Comments and Suggestions for Authors

The authors report on a retrospective study on 899 patients diagnosed with MCC in Canada in the years 2000 – 2018. Data was retrieved from relevant sources across centers and stored in a dedicated database. The authors compared survival of the Canadian patients with the American cohort reported in the AJCC 8th edition. Primary outcome was 5 year OS, and secondary outcome 5 year DFS and CSS.

The study is well presented including considerations of discrepancies between the two cohorts, limitations of the study and the manifestation that the outcome serves as adding to the baseline data of the “pre immunotherapy” era. Still, there are some points the authors should address:

  • One would gather that the source of data used is the same as published by Megan Delisle et al Ann Surg Oncol 2025? If so, there is a discrepancy since there is no mentioning of chemotherapy alone or in combination with radiotherapy in this study as previously published by Delisle et al.
  • If chemotherapy was administered this should be included and depicted in Treatment Received.
  • Is there information on the chemotherapy regimens used (eg carboplatin + etoposide etc)
  • What impact could chemotherapy have on the differences found between the two cohorts
  • What treatments were provided upon recurrence: equally between the two cohorts, type of systemic treatments? This could impact OS, CSS significantly and should be taken into consideration in the paper.

Reviewer 2 Report

Comments and Suggestions for Authors

the authors reported a multicenter retrospective studies on Merkel  on 899 patients (36.4%  stage I, 21.7%stage II,33.9%stage III, 8.0%stage IV ) mostly male and 10.2immunosuppressed, and age at diagnosis was 75 years.

The authors reported OS by stage and found it was lower for patients diagnosed in Canada , although it must be stressed that the cohort considered the period 2000-2018 and therefore without immunotherapy implementation.

Indeed Zaggana E ( Cancers 2022) reported a 5 years OS of 64%

In particular, the authors reported a  lower survival rates in IIIb compared with the AJCC 8th edition survival rates. Moreover, they found patients with regional lymph node 
macrometastases and an unknown primary (N1b unknown primary) had similar survival
to patients diagnosed with microscopic regional lymph node metastases (N1a).

Although Merkel cell carcinoma review and retrospectives studies have been published the article might be of interest to readers

References are adequate

Minor revisions:

in table 1 please change site with hospital to avoid confusion

provide p value in a different column

In table 2 correct typing TAny separate T and any 

Comments on the Quality of English Language

English is fine
